# UV Radiation Effect in New Materials Developed for the Construction of Beehives

**DOI:** 10.3390/polym15214249

**Published:** 2023-10-28

**Authors:** Andrés Rubiano-Navarrete, Camilo Lesmes Fabian, Yolanda Torres-Pérez, Edwin Gómez-Pachón

**Affiliations:** 1Grupo de Investigación DITMAV—Diseño, Innovación y Asistencia Técnica para Materiales Avanzados, Joven Investigador, Universidad Pedagógica y Tecnológica de Colombia-UPTC, Tunja 150003, Colombia; 2Dirección de Investigaciones, Corporación Universitaria del Meta, UniMeta, Villavicencio 500001, Colombia; camilo.lesmes@unimeta.edu.co; 3Grupo de Investigación en Energía y Nuevas Tecnologías—GENTE, Escuela Ingeniería Electromecánica, Facultad Duitama, Universidad Pedagógica y Tecnológica de Colombia—UPTC, Duitama 150461, Colombia; yolanda.torres01@uptc.edu.co; 4Grupo de Investigación DITMAV—Diseño, Innovación y Asistencia Técnica para Materiales Avanzados, Escuela de Diseño Industrial, Universidad Pedagógica y Tecnológica de Colombia-UPTC, Duitama 150461, Colombia; edwin.gomez02@uptc.edu.co

**Keywords:** beehives, composite materials, UV radiation, accelerated degradation, HDPE, polymers

## Abstract

In recent decades, there has been an increasing focus on the alarming decline in global bee populations, given their critical ecological contributions to natural pollination and biodiversity. This decline, marked by a substantial reduction in bee colonies in forested areas, has serious implications for sustainable beekeeping practices and poses a broader risk to ecological well-being. Addressing these pressing issues requires innovative solutions, one of which involves the development and fabrication of beehives crafted from composite materials that are ecologically compatible with bee biology. Importantly, these materials should also exhibit a high resistance to environmental factors, such as ultraviolet (UV) radiation, in order to maintain their mechanical integrity and longevity. To investigate this, we conducted accelerated UV degradation tests on a variety of composite materials to rapidly assess their susceptibility to UV-induced changes. High-density polyethylene (HDPE) served as the matrix material and was reinforced with natural fibers, specifically fique fibers (*Furcraea bedinghausii*), banana fibers, and goose feathers. Our findings indicate that UV radiation exposure results in a noticeable reduction in the tensile strength of these materials. For example, wood composites experienced a 48% decline in tensile strength over a 60-day period, a rate of deterioration notably higher than that of other tested composite materials. Conversely, HDPE composites fortified with banana fibers initially demonstrated tensile strengths exceeding 9 MPa and 10 MPa. Although these values gradually decreased over the observation period, the composites still displayed favorable stress–strain characteristics. This research underscores the substantial influence of UV radiation on the longevity and efficacy of beehive materials, which in turn affects the durability of natural wood hives exposed to these environmental factors. The resultant increased maintenance and replacement costs for beekeepers further emphasize the need for judicious material selection in beehive construction and point to the viability of the composite materials examined in this study.

## 1. Introduction

Over recent decades, one of the primary concerns in the loss of global biodiversity that has been evidenced by the scientific community is the declining population of pollinating bees. These are one of the groups of insects that have a direct impact on various environmental processes and ecosystem stability. Natural pollination by bees influences the processes of seed production, fruiting, natural regeneration, and the renewal of many plant species. For this reason, they are key species at the ecosystem level and for the maintenance of natural reserves [1,2]. The mutualistic relationship formed from the interaction of pollinating bees with plants promotes biodiversity, richness, genetic variability, and sustenance of a large number of flora and fauna species worldwide. This contributes to their propagation and food security. Consequently, bees have been cataloged as one of the world’s most valuable species. Without them, a chain would collapse, impacting other species dramatically [1,3,4]

Moreover, from an anthropological standpoint, bees have a significant impact on human activities, ranging from food security and agriculture to economic stability. This is achieved through the commercialization of bee products like honey, propolis, wax, royal jelly, and other by-products. These are used in fields such as medicine, industry, cosmetics, and as part of beekeeping families’ economies [2,5].

The massive loss of bee colonies, also known as the Colony Collapse Disorder, is a phenomenon stemming from various factors. It threatens the natural pollination process and, in turn, ecosystem stability [3,6]. While there are multiple negative factors impacting bee populations, such as toxic pesticides and climate change [7,8], studies emphasize that environmental conditions inside hives directly affect bee well-being. One such example is the hive’s internal temperature [9,10]. Similarly, the materials used for hive construction are a key for bees’ requirements and acceptance [7]. Throughout human history, in the realm of beekeeping, various techniques and hive construction materials have been developed. These include clay, logs, cork, straw, plant fibers, cement, and others that have evolved with the technique to improve colony quality and stability [11,12]. Currently, numerous designs and materials have been used for hive construction, with wooden hives being the most developed and adapted to bees’ biological processes [13]. Additionally, wood contributes to hive thermoregulation and is easy to design and repair, making it a preferred material for hive fabrication [14]. However, wood’s lifespan depends on its quality. Some wood may last just a few months, while others can endure for years, making it the most critical parameter for achieving hive sustainability [7,11].

While wood is affected by various factors, such as dew, rain, snow, and wind [8,11], natural ultraviolet (UV) light is the primary cause of deterioration for beekeeping construction materials, especially wood. Studies by Rodríguez and Fuentes [8] found that wood changes color after being exposed to UV light for a month. It loses surface properties, leading to the appearance of cracks and loss of uniformity and mechanical strength, as well as the accelerated decomposition of cellulose and lignin. This deterioration can also foster wood-decaying fungi, particularly in damp areas [15,16]. Due to this, beekeepers are exploring other materials, such as high-density polyethylene (HDPE) [7]. While HDPE is widely used in construction, when utilized outdoors, UV light increases photo-oxidation, chemical crystallization, and brittleness, reducing mechanical properties like ductility due to the loss in molecular weight from the formation of carbonyl groups [17,18]. HDPE’s exposure to UV can cause the degradation of its additives and stabilizers, which are used to enhance its heat and light resistance, making it fragile and brittle. Consequently, to enhance its longevity, this material necessitates integration with other organic substrates, with a continuous emphasis on ensuring compatibility with the biological requirements of bees in hive construction [7,18].

This article aims to explore new materials for constructing beehives by studying the effects of UV radiation on mechanical properties. For this, a device for accelerated degradation was constructed, enabling a quick evaluation of UV light effects and shortening the study’s duration.

## 2. Materials and Methods

### 2.1. Materials Combination Acquisition

In this experiment, we worked with recycled high-density polyethylene obtained from the company ZEROWASTE. This material was combined with organic products to enhance the bees’ acceptance and improve the material’s durability. The organic products were fique fibers (*Furcraea bedinghausii*), pseudostem banana fibers, and goose feathers. The pseudostem banana fibers were obtained from the municipality of Moniquirá, Boyacá, which is located 1669 m above sea level and at an ambient temperature of 20 °C. The fique fibers and goose feathers were acquired from the municipality of Duitama, Boyacá, which is located 2509 m above sea level and at an ambient temperature of 15 °C. For the experimental work, the banana and fique fibers were manually separated and cut to an average length of 5–6 mm. The cut fibers were immersed in 5% NaOH for 12 h, then rinsed with distilled water, and the fibers were bleached using 30% hydrogen peroxide. They were again washed and sun-dried to remove impurities from the fiber surface. The material was homogenized in terms of particle size after drying, using a sieve process with a 1.0 mm sieve. Finally, the fibers were dried in a hot air oven at 80 °C for 5 h to remove moisture and were stored in a sealed bag for later processing. Regarding the goose feathers, they were crushed to produce particles 5–6 mm in diameter. These were then treated with 2% alkali for 4 h, sun-dried, and further dried in an oven at 70 °C for 5 h.

### 2.2. Specimen Manufacturing

The molding process was carried out by hot pressing. An extrusion process was used to mix the polymer and fibers, aiming to achieve better fiber distribution within the polymer matrix. Three different materials were created with distinct natural fiber reinforcements (i.e., banana, fique, goose feathers) at two different weight percentages (10% and 15%, respectively). These materials were fed through a hopper and mixed at a speed of 50 rpm and a temperature of 170 °C. Composite filaments were produced from the extruder and passed through a container of cold water. These strands were combined to produce granules averaging 4–5 mm in length using a pelletizer. These granules were kept in a hot air oven at 90 °C for 5 h to remove moisture. Subsequently, the granules underwent a hot pressing process in a mold of 18 cm × 14 cm, adding 80 g of composite material and pressing at 170 °C to achieve a thickness of 3 mm. The cooling time was 2 h. The mold was designed so that, in a single pressing process, specimens could be manufactured for the accelerated UV radiation degradation test and the tensile stress test according to ASTM D638 standard. A total of 6 replicas were made per material, with 3 replicas of each material exposed for 30 days and the other 3 replicas for 60 days. Considering that 7 materials were evaluated, 42 specimens were manufactured in total. Table 1 summarizes the evaluated composite materials.

### 2.3. Accelerated Degradation by Ultraviolet Radiation

For the UV accelerated degradation test, the first task undertaken was to dry each of the seven materials in a hot air drying oven until reaching a constant weight. On the other hand, a sealed chamber was constructed to prevent UV light emissions from escaping. Short-wave UV lamps were used, with these made from a phosphor-free fluorescent lamp tube, composed of fused quartz, as regular glass absorbs UV-C. In the UV radiation chamber, a lamp was used that emits ultraviolet light, with two peaks in the UV-C band at 253.7 nm and 185 nm. The fused quartz tube allows the passage of the 253.7 nm radiation but blocks the 185 nm wavelength. These low-pressure tubes have a UV-C output two or three times higher than a regular fluorescent lamp tube. These lamps have a typical efficiency of approximately 30–40%, meaning that for every 100 W (watts) of electricity consumed by the lamp, they will produce approximately 30–40 W of total UV output. They also emit a bluish–white visible light due to the other spectral lines of mercury. The specimens were placed inside the chamber at a distance of 25–30 cm from the UV light lamp. The accelerated degradation test due to UV light exposure was conducted in the laboratories of the Santo Tomás University, in the city of Tunja, Colombia.

### 2.4. Tensile Stress Test

The tensile test was carried out on the specimens of the developed composite materials, considering the Colombian Standard ASTM D638, which defines the “definitive guide for tensile tests of plastics”. The test was conducted at three UV exposure times: 0, 30, and 60 days. Exposure was continuous, both day and night. After 30 days, 3 of the replicas were removed and taken to the laboratory for tensile testing. This test was conducted in the laboratories of the Pedagogical and Technological University of Colombia, Duitama campus. The tensile test was performed on the universal test machine WDW 100, UTM, at a crosshead speed of 5 mm/min, for all the specimens to be evaluated.

### 2.5. Statistical Analysis

A variance analysis was carried out for the factors “Fiber Type” and “Fiber Quantity%”. This was performed for the 18 cases. The procedure aimed to determine the statistically significant effect of UV radiation on the various materials obtained. Likewise, multiple range tests were performed to determine significant differences between all the means.

## 3. Results

### 3.1. Tensile Strength Test

Figure 1 and Table 2 display the tension results obtained from the universal testing machine in each of the tests with their respective UV-C radiation exposure times (0, 30, and 60 days). From there, the degradation of the tensile strength of the different materials can be observed, where it was found that wood, being the reference material, is the most resistant to tension. Similarly, material F based on HDPE and 15% fique is the most resistant material after 60 days of accelerated degradation. It is also worth noting that material E based on HDPE and 10% fique does not show significant variations in its tensile strength after 60 days of exposure.

Figure 2 shows the samples of the developed composite materials exposed to ultraviolet radiation. After their respective exposure to UV radiation, each of these samples is subjected to mechanical stress resistance testing by means of the universal testing machine as shown in Figure 3.

Figure 4 shows how ultraviolent radiation affects the tensile strength of each of the composite materials developed.

The longer polymeric materials are exposed to ultraviolet radiation, the more their tensile strength will decrease; however, it is more noticeable in some materials than others. As for the wood, it can be observed that when there is no exposure to ultraviolet radiation, a certain percentage of moisture is retained, which makes it somewhat flexible. On the contrary, when exposed to ultraviolet radiation, there is a noticeable decrease in the deformation of the sample. This is because ultraviolet radiation tends to remove the moisture content from the sample, causing the wood to become dry and brittle. As can be seen in Figure 1, the wood’s decrease in tensile strength is significant compared to the developed materials.

### 3.2. Statistical Analysis Results

As can be seen in Table 3, the results of the statistical analysis show that the *p*-values are very low, indicating significant differences both for main effects and for interactions. That is, the tensile strength was high for the material based on pine wood, considering it as the traditional material in hive construction. Meanwhile, the material based on HDPE and 15% fique fiber is the composite material with the highest tensile strength.

Effect of the fique fiber: According to Figure 5, there is a significant difference between all composite materials, with wood showing the highest tensile strength. Among all HDPE materials, the mix with fique fiber shows the highest tensile strength.

Effect of banana fiber: Previous studies [19] developed a virgin HDPE material with 20% banana fiber, whose tensile strength decreased by 19% compared to virgin HDPE samples. Likewise, the tensile strength of recycled HDPE compounds reinforced with 20% banana fiber decreased by 3.21% compared to unreinforced recycled HDPE samples. This decrease in strength is due to the incorporation of short natural fibers, which promote multiple nucleation and reduce the formation of long polymeric chains. Comparing these published results with the obtained results confirms that the tensile strength of the different materials evaluated is inversely proportional to the amount of short banana fiber in the material. That is, the higher the banana fiber content, the lower the tensile strength the composite material will have. Figure 6 shows the fracture caused by the traction of the recycled high-density polyethylene composite reinforced with 10% banana fiber, where the adhesion between the matrix and the reinforcement is observed.

When polymeric materials are constantly exposed to ultraviolet light, they lose mechanical properties, evidenced by a decrease in tensile resistance. In the case of wood, moisture plays a significant role in both the strength and deformation of samples, depending on moisture loss, a factor that increases the flexibility of this material. It is noteworthy that due to moisture loss, materials showed a slight buckling after the exposure time.

In Figure 7, the hive developed with recycled high-density polyethylene reinforced with 15% banana fiber is observed, because it was the composite material that highlighted its behavior within the parameters stable not only to resistance to ultraviolet radiation but also to other factors such as humidity, fungal proliferation, and thermal conductivity.

## 4. Discussion

Within the scope of this study, a comprehensive assessment of tensile strength has been conducted on various polymeric materials exposed to ultraviolet radiation and reinforced with natural fibers, aiming to elucidate the influence of radiation and fiber incorporation on the mechanical properties of the resultant composites. The outcomes demonstrate a gradual reduction in tensile strength as the polymeric materials undergo extended exposure to ultraviolet radiation. While this trend is observable across all materials, it is noteworthy that certain materials exhibit a more pronounced response, thereby revealing a distinct sensitivity to accelerated degradation conditions [4,10].

Of particular interest is the observation regarding wood, which unveils an inversely proportional relationship between tensile strength and ultraviolet radiation exposure. This phenomenon is attributed to the radiation capability to eliminate moisture from wood, leading to heightened fragility and a significant decline in its strength. In contrast, HDPE and fique fiber-based composite materials exhibit a contrasting behavior. Material F, comprising 15% fique fiber, emerges as the most resilient after 60 days of degradation, suggesting that the synergy between HDPE and an appropriate fique fiber content can confer enhancement in tensile strength, likely due to an augmented interaction between the polymeric matrix and the fibers [7,8,20].

Furthermore, the incorporation of natural fibers, such as banana fiber, is observed to induce a reduction in tensile strength of the composite materials, attributed to the presence of short fibers that facilitate the formation of multiple nucleation points and a reduction in continuous polymeric chains. The findings of this study not only enrich our understanding of the influence of ultraviolet radiation on polymeric materials but also provide valuable insights into the interplay between natural fibers and polymeric matrices in terms of mechanical properties. These results support the notion that the tensile strength of composite materials is directly correlated with the quantity and nature of incorporated fibers [3,9,15,16].

## 5. Conclusions

After the developed materials were exposed to ultraviolet radiation and their tensile strength was evaluated at 0, 30, and 60 days of exposure, it was found that there was a decrease in the tensile strength of each of the materials, by approximately 2 MPa. However, the effect of ultraviolet radiation on wood decreases over the 60 days of exposure by 48.43% less than its initial tensile strength. Nevertheless, even though wood has a tensile strength of 29.42 MPa and composite materials have a tensile strength no greater than 11 MPa, wood significantly decreases its tensile strength when exposed to ultraviolet radiation compared to the developed composite materials. This indicates that these composite materials have a greater resistance to ultraviolet radiation degradation than pine wood.

It is evident that the developed composite materials of high-density polyethylene reinforced with 10% and 15% banana fiber have a tensile strength on day 0 of 10.38 MPa and 9.55 MPa, respectively. The composite material reinforced with banana fiber is the most tensile resistant compared to the fique and goose feather reinforcements. For the composite material developed from high-density polyethylene reinforced with 10% banana fiber, it was observed that after the first 30 days of exposure to ultraviolet radiation, it decreased from 10.38 MPa to 8.88 MPa. However, even though there is a decrease of 1.5 MPa, the sample exposed to 60 days of exposure resulted in a tensile strength of 8.58 MPa, indicating a decrease of 0.3 MPa.

This means that from day 0 to day 30, the tensile strength decreased by 1.5 MPa, but from day 30 to day 60, there was a decrease of 0.3 MPa. This demonstrates that the composite material, when exposed to ultraviolet radiation, decreases its tensile strength property. Although pine wood presents a high tensile strength, it shows low resistance to ultraviolet degradation due to its decrease in mechanical properties.

The prospect of using composite materials against ultraviolet degradation in beekeeping applications is promising and offers significant advantages. From the studies carried out, we can finalize the following:

Composite materials developed from agro-industrial waste, despite not having a high tensile strength when exposed to ultraviolet radiation, decrease their mechanical properties minimally compared to the wood sample, which significantly decreases their mechanical strength. This demonstrates that recycled HDPE reinforced with natural fibers has greater durability compared to traditional materials used in beekeeping, such as wood. This reduces the need to constantly replace or maintain components of hives and other beekeeping equipment.

By using composite materials that last longer, the amount of waste and the need to cut down trees or other natural resources for the construction of hives and other beekeeping accessories are reduced. This has a positive environmental impact by decreasing the ecological footprint of beekeeping.

The resistance of composite materials to ultraviolet degradation also contributes to maintaining a stable environment for bees. Hives and equipment that remain in good condition for longer provide a more reliable and comfortable shelter for bees, which can result in more efficient and healthy honey production.

Despite a slightly higher initial investment in composite materials, the long-term benefits outweigh this cost. The reduced need for replacement and constant maintenance, as well as improved efficiency in beekeeping production, can translate into significant savings for beekeepers.

The perspective of the use of composite materials in beekeeping applications against ultraviolet degradation is highly positive. These materials offer a durable, environmentally friendly, and efficient solution that benefits both beekeepers and bees, thus promoting sustainability in the beekeeping industry.

This future work directly impacts determining the material’s lifespan under conditions exposed to ultraviolet radiation, since ultraviolet radiation is one of the factors that causes deterioration in the physical properties of materials. This forces beekeepers to change hives, leading to higher costs in production units.

## Figures and Tables

**Figure 1 polymers-15-04249-f001:**
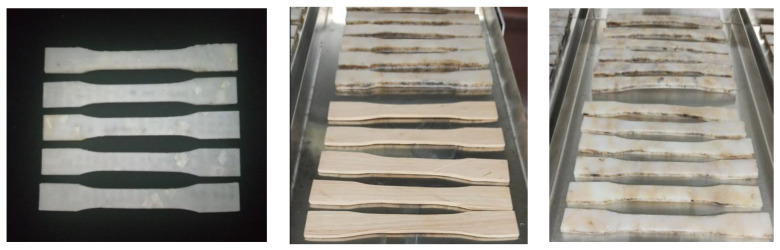
Composite material samples before undergoing accelerated UV degradation testing.

**Figure 2 polymers-15-04249-f002:**
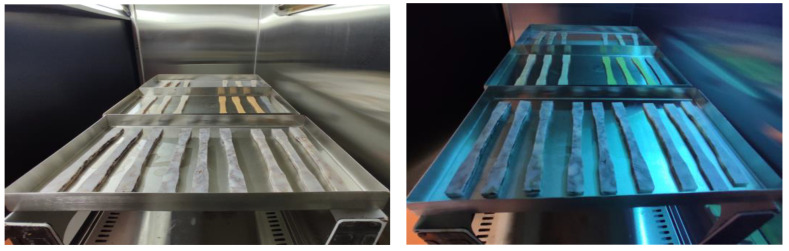
Samples exposed in the UV radiation chamber before and after the lamp is turned on.

**Figure 3 polymers-15-04249-f003:**
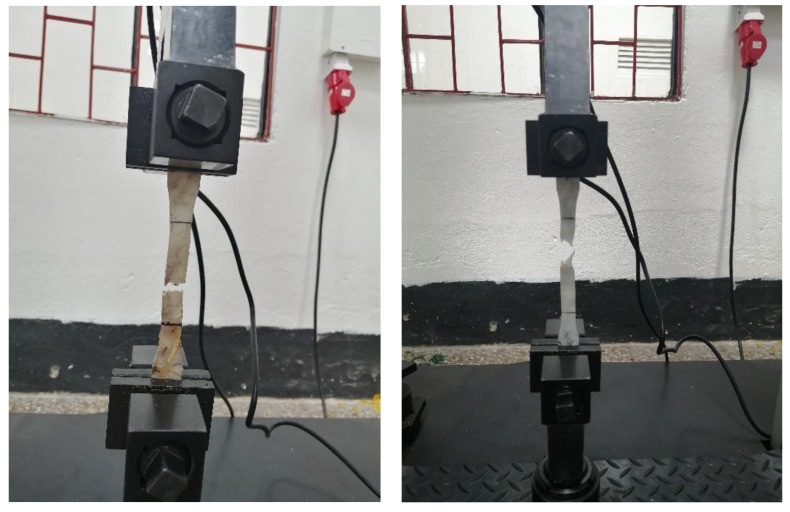
Universal testing machine for tensile tests with specimens before and after rupture, within the range established according to ASTM D638 standard.

**Figure 4 polymers-15-04249-f004:**
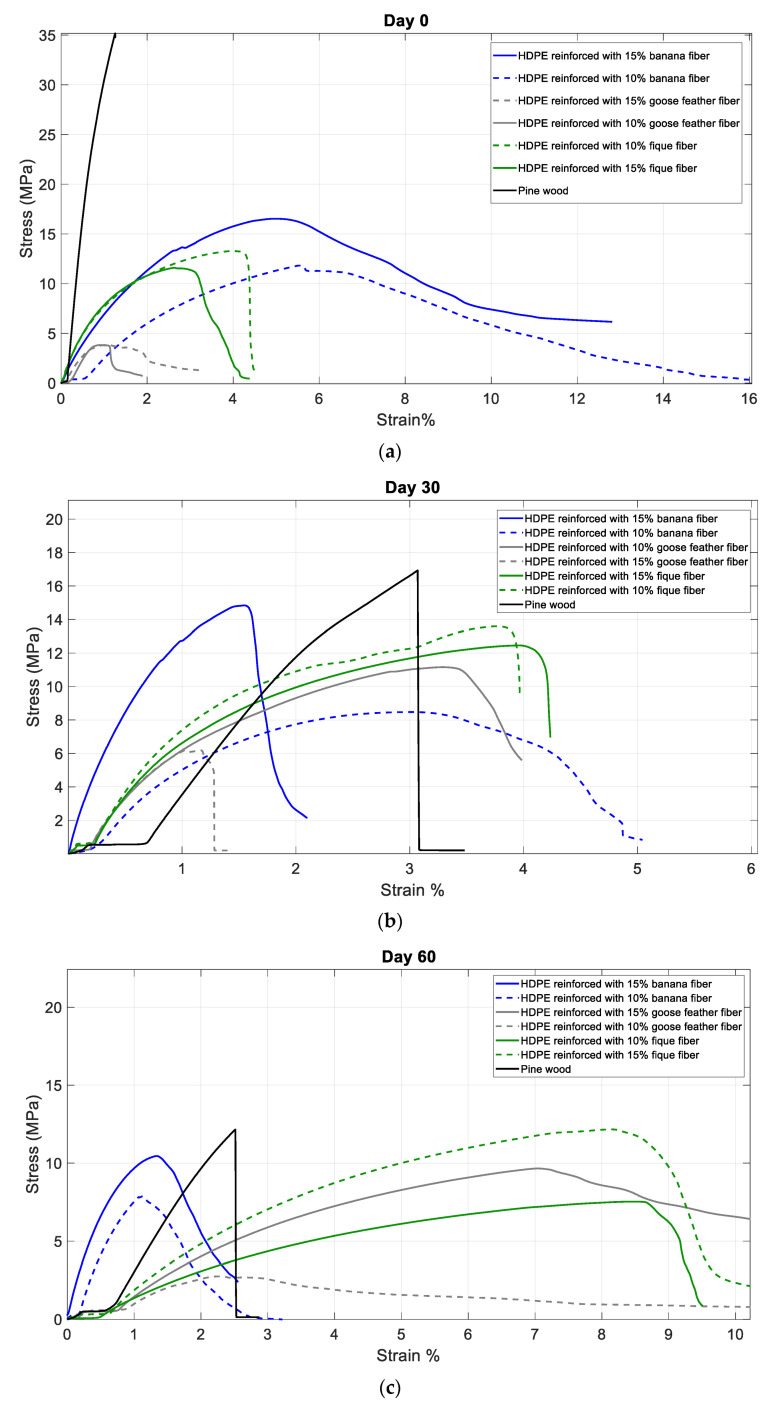
Tensile curves of the different samples at different days of exposure to ultraviolet radiation: (**a**) Day 0, (**b**) Day 30, and (**c**) Day 60.

**Figure 5 polymers-15-04249-f005:**
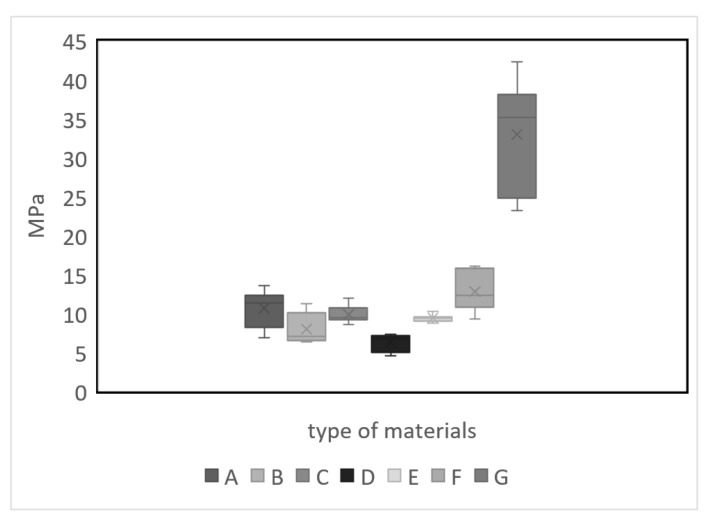
Multifactorial ANOVA means graph—UV Radiation.

**Figure 6 polymers-15-04249-f006:**
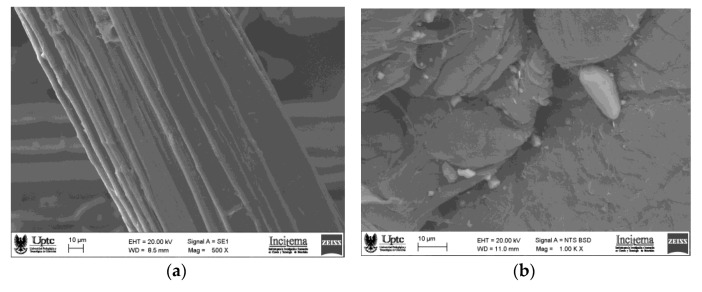
(**a**) Micrograph of banana fiber, (**b**) micrograph of recycled high-density polyethylene, (**c**) adhesion between the polymer matrix and the banana fiber, (**d**) micrograph of the fiber immersed in the polymer matrix, and (**e**) fracture of the sample of recycled high-density polyethylene reinforced with 10% banana fiber by weight.

**Figure 7 polymers-15-04249-f007:**
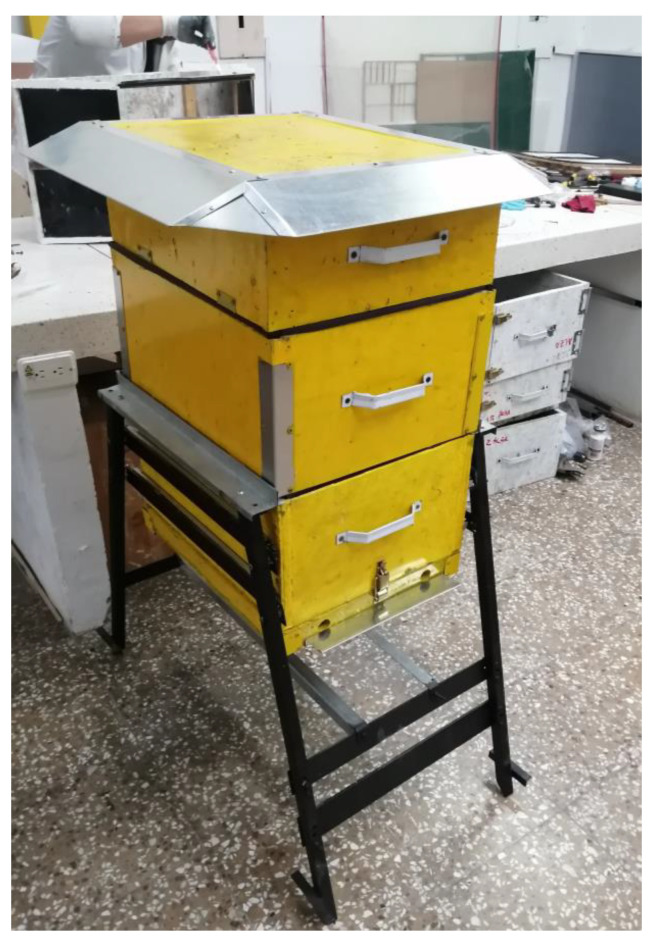
Manufacture of beehive with recycled high-density polyethylene composite material reinforced with 15% banana fiber.

**Table 1 polymers-15-04249-t001:** Nomenclature of different materials and their compositions.

Nomenclature	Material
A	HDPE reinforced with 10% banana fiber
B	HDPE reinforced with 15% banana fiber
C	HDPE reinforced with 10% goose feather fiber
D	HDPE reinforced with 15% goose feather fiber
E	HDPE reinforced with 10% fique fiber
F	HDPE reinforced with 15% fique fiber
G	Pine wood

**Table 2 polymers-15-04249-t002:** Results of the tensile strength test for the materials evaluated at 0, 30, and 60 days of continuous UV light exposure.

	A	B	C	D	E	F	G
Day 0	12.4 ± 1.2	10.5 ± 0.7	11.2 ± 0.8	7.2 ± 0.1	9.5 ± 0.7	15.9 ± 0.2	39.5 ± 2.4
Day 30	11.8 ± 0.5	6.9 ± 0.4	9.5 ± 0.1	6.7 ± 0.5	9.4 ± 0.3	12.3 ± 0.4	35.2 ± 1.1
Day 60	7.8 ± 0.8	6.6 ± 0.2	9.1 ± 0.5	4.9 ± 0.4	9.2 ± 0.4	10.3 ± 0.9	24.3 ± 1.9

**Table 3 polymers-15-04249-t003:** Analysis of Variance for UV Radiation—Type III Sum of Squares.

Source	Sum of Squares	Gl	Mean Squared	F-Ratio	*p*-Value
A: Type of Fiber	41.264	2	20.632	98.19	0.0000
B: % Quantity	26.645	1	26.645	126.81	0.0000
Interactions					
AB	5.6497	2	2.82485	13.44	0.0009
Residuals	2.5214	12	0.210117		
Total (Corrected)	76.0801	17			

## Data Availability

The data presented in this study are available on request from the corresponding author.

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
