# Peer review of "UV Radiation Effect in New Materials Developed for the Construction of Beehives"

_polymers, 2023, doi:10.3390/polym15214249_

Round 1

Reviewer 1 Report

1. The abstract should be refined

2. Why do you explain more about the beehives? Actually, the influence of UV radiation on mechanical properties of composite materials based on HDPE and fiber was only investigated.

3. The novelty and technological level should be improved

4. SEM of the different specimens before and after rupture should be provided

5. The tensile curve of the different specimens should be provided for convenient comparison

Moderate editing of English language required

Author Response

Please seen the attachment.

Reviewer 2 Report

Review on polymers-2582536-peer-review-v1, “UV Radiation Effect in New Materials Developed for the Construction of Beehives”. By Rubiano-Navarrete et al.

In this work, HDPE-natural fiber composites were studied, where the natural fibers were goose feather and fique and banana fibers. The fibers and matrix were mixed by extrusion and specimens were obtained by compression molding. The composites were subjected to ultraviolet radiation for determined periods, the tensile strength was evaluated, and the fracture morphology was observed. The study is elementary and its contribution to the design of materials for beekeeping use is not very prominent. In the present form, the publication of this manuscript in Polymers is not recommended.

 Comments

a) English, although it is understandable, requires a deep revision.

b) According to the manuscript, the purpose of this study is to propose new materials for beekeeping use; however, in the end, a valuable result for the application is not appreciated.

c) In most of the studies focused on the effect of ultraviolet radiation on polymeric materials, if not all, the decrease in mechanical properties is evident. From this point of view, what is the contribution of this study to the subject in question?

d) An FTIR study was not carried out to analyze the evolution of the carbonyl group.

e) An alkaline treatment was given to the fibers before mixing with the matrix; however, there is no mention of the possible effect of said treatment.

f) In Table 1, composites E and F require revision.

g) In the results, it is commented that the fibers are distributed homogeneously in the matrix, but Figure 1, Figure 2, and Figure 6 clearly show that the fibers are agglomerated. This affects the mechanical properties as well as UV radiation.

h) Lines 215 to 218 mention a recycled HDPE - banana fiber composite, but a reference is not indicated.

i) Line 222 refers to Figure 6 (also line 226). It is mentioned that the figure describes a recycled HDPE composite with 10% banana fiber. However, Section 2, Materials and Methods, line 98, speaks of a virgin HDPE. Where did the recycled HDPE come from?

j) Why is an emphasis placed on the places where the studies were carried out? This is irrelevant information unless some particular environmental condition warrants such mentions.

k) Table 2 and Figure 4 provide the same information. One of them should be removed from the manuscript.

l) Although the pine wood specimens were taken as "the reference material," it is necessary to include virgin HDPE specimens as the "blank" to determine the effect of UV radiation on the matrix alone. In this way, the differences observed in the tensile strength in the composites could be attributed to the interactions of each type of fiber on the thermoplastic matrix. It could also be determined, if it exists, whether or not the type of fiber exerts some protection to the matrix against UV radiation.

m) The manuscript minimally mentions the composite reinforced or loaded with goose feather fiber.

n) Mention is made of the effect of UV radiation on the pine wood probes, which is related to the absorbed humidity. Usually, before this type of study, all materials under evaluation should undergo a heat treatment to eliminate moisture as much as possible.

o) In lines 254-257, it is mentioned that the presence of short banana fibers affects the continuity of the polymeric chains. What is the meaning of this statement? Does it mean that the fibers degrade the matrix?

p) The conclusions are a summary of the results. Although some authors like this type of format, a conclusion about the objective of the work has not been issued. That is, what is the perspective of the use of the composite materials studied in beekeeping applications?

English, although it is understandable, requires a deep revision.

(I clarify that my level of English also requires improvement.)

Round 2

Reviewer 1 Report

this manuscript can be accepted in present form

this manuscript can be accepted in present form

Author Response

Please seen the attachment.
